# SENSEI: SENSITIVE SET INVARIANCE FOR ENFORCING INDIVIDUAL FAIRNESS

**Mikhail Yurochkin**
IBM Research
MIT-IBM Watson AI Lab
mikhail.yurochkin@ibm.com

**Yuekai Sun**
Department of Statistics
University of Michigan
yuekai@umich.edu

## ABSTRACT

In this paper, we cast fair machine learning as invariant machine learning. We first formulate a version of individual fairness that enforces invariance on certain sensitive sets. We then design a transport-based regularizer that enforces this version of individual fairness and develop an algorithm to minimize the regularizer efficiently. Our theoretical results guarantee the proposed approach trains certifiably fair ML models. Finally, in the experimental studies we demonstrate improved fairness metrics in comparison to several recent fair training procedures on three ML tasks that are susceptible to algorithmic bias.

## 1 INTRODUCTION

As machine learning (ML) models replace humans in high-stakes decision-making and decision-support roles, concern regarding the consequences of algorithmic bias is growing. For example, ML models are routinely used in criminal justice and welfare to supplement humans, but they may have racial, class, or geographic biases (Metz & Satariano, 2020). In response, researchers proposed many formal definitions of algorithmic fairness as a first step towards combating algorithmic bias.

Broadly speaking, there are two kinds of definitions of algorithmic fairness: group fairness and individual fairness. In this paper, we focus on enforcing individual fairness. At a high-level, the idea of individual fairness is the requirement that a fair algorithm should treat similar individuals similarly. Individual fairness was dismissed as impractical because there is no consensus on which users are similar for many ML tasks. Fortunately, there is a flurry of recent work that addresses this issue (Ilvento, 2019; Wang et al., 2019; Yurochkin et al., 2020; Mukherjee et al., 2020). In this paper, we assume there is a similarity metric for the ML task at hand and consider the task of enforcing individual fairness. Our main contributions are:

1. we define *distributional individual fairness*, a variant of Dwork et al.'s original definition of individual fairness that is (i) more amenable to statistical analysis and (ii) easier to enforce by regularization;
2. we develop a stochastic approximation algorithm to enforce distributional individual fairness when training smooth ML models;
3. we show that the stochastic approximation algorithm converges and the trained ML model generalizes under standard conditions;
4. we demonstrate the efficacy of the approach on three ML tasks that are susceptible to algorithmic bias: income-level classification, occupation prediction, and toxic comment detection.

## 2 ENFORCING INDIVIDUAL FAIRNESS WITH SENSITIVE SET INVARIANCE (SENSEI)

### 2.1 A TRANSPORT-BASED DEFINITION OF INDIVIDUAL FAIRNESS

Let $\mathcal{X}$ and $\mathcal{Y}$ be the space of inputs and outputs respectively for the supervised learning task at hand. For example, in classification tasks, $\mathcal{Y}$ may be the probability simplex. An ML model is a function $h : \mathcal{X} \to \mathcal{Y}$ in a space of functions $\mathcal{H}$ (*e.g.* the set of all neural nets with a certain architecture).

Dwork et al. (2011) define *individual fairness* as $L$-Lipschitz continuity of an ML model $h$ with respect to appropriate metrics on $\mathcal{X}$ and $\mathcal{Y}$:

$$d_{\mathcal{Y}}(h(x), h(x')) \leq L d_{\mathcal{X}}(x, x') \tag{2.1}$$

for all $x, x' \in \mathcal{X}$. The choice of $d_{\mathcal{Y}}$ is often determined by the form of the output. For example, if the ML model outputs a vector of the logits, then we may pick the Euclidean norm as $d_{\mathcal{Y}}$ (Kannan et al., 2018; Garg et al., 2018). The metric $d_{\mathcal{X}}$ is the crux of (2.1) because it encodes our intuition of which inputs are similar for the ML task at hand. For example, in natural language processing tasks, $d_{\mathcal{X}}$ may be a metric on word/sentence embeddings that ignores variation in certain sensitive directions. In light of the importance of the similarity metric in (2.1) to enforcing individual fairness, there is also a line of work on learning the similarity metric from data (Ilvento, 2019; Wang et al., 2019; Mukherjee et al., 2020). In our experiments, we adapt the methods from Yurochkin et al. (2020) to learn similarity metrics.

Although intuitive, individual fairness is statistically and computationally intractable. Statistically, it is generally impossible to detect violations of individual fairness on zero measure subset of the sample space. Computationally, individual fairness is a Lipschitz restriction, and such restrictions are hard to enforce. In this paper, we address both issues by lifting (2.1) to the space of probability distributions on $\mathcal{X}$ to obtain an "average case" version of individual fairness. This version (i) is more amenable to statistical analysis, (ii) is easy to enforce by minimizing a data-dependent regularizer, and (iii) preserves the intuition behind Dwork et al. (2011)'s original definition of individual fairness.

**Definition 2.1** (distributional individual fairness (DIF)). *Let $\epsilon, \delta > 0$ be tolerance parameters and $\Delta(\mathcal{X} \times \mathcal{X})$ be the set of probability measures on $\mathcal{X} \times \mathcal{X}$. Define*

$$R(h) \triangleq \left\{ \begin{array}{ll} \sup_{\Pi \in \Delta(\mathcal{X} \times \mathcal{X})} & \mathbf{E}_{\Pi}\big[d_{\mathcal{Y}}(h(X), h(X'))\big] \\ \textit{subject to} & \mathbf{E}_{\Pi}\big[d_{\mathcal{X}}(X, X')\big] \leq \epsilon \\ & \Pi(\cdot, \mathcal{X}) = P_X \end{array} \right\}. \tag{2.2}$$

*where $P_X$ is the (marginal) distribution of the inputs in the ML task at hand. An ML model $h$ is $(\epsilon, \delta)$-distributionally individually fair (DIF) iff $R(h) \leq \delta$.*

We remark that DIF only depends on $h$ and $P_X$. It does not depend on the (conditional) distribution of the labels $P_{Y|X}$, so it does not depend on the performance of the ML model. In other words, it is possible for a model to perform poorly and be perfectly DIF (*e.g.* the constant model $h(x) = 0$).

The optimization problem in (2.2) formalizes *correspondence studies* in the empirical literature (Bertrand & Duflo, 2016). Here is a prominent example.

**Example 2.2.** *Bertrand & Mullainathan studied racial bias in the US labor market. The investigators responded to help-wanted ads in Boston and Chicago newspapers with fictitious resumes. To manipulate the perception of race, they randomly assigned African-American or white sounding names to the resumes. The investigators concluded there is discrimination against African-Americans because the resumes assigned white names received 50% more callbacks for interviews than the resumes.*

*We view Bertrand & Mullainathan's investigation as evaluating the objective in (2.3) at a special $T$. Let $\mathcal{X}$ be the space of resumes, and $h : \mathcal{X} \to \{0, 1\}$ be the decision rule that decides whether a resume receives a callback. Bertrand & Mullainathan implicitly pick the $T$ that reassigns the name on a resume from an African-American sounding name to a white one (or vice versa) and measures discrimination with the difference between callback rates before and after reassignment:*

$$\mathbf{E}_P\big[1\{h(X) \neq h(T(X))\}\big] = \mathbf{P}\{h(X) \neq h(T(X))\}.$$

We consider distributional individual fairness a variant of Dwork et al.'s original definition. It is not a fundamentally new definition of algorithmic fairness because it encodes the same intuition as Dwork et al.'s original definition. Most importantly, it remains an individual notion of algorithmic fairness because it compares individuals to similar (close in $d_{\mathcal{X}}$) individuals.

## 2.2 DIF VERSUS INDIVIDUAL FAIRNESS

It is hard to directly compare Dwork et al.'s original definition of individual fairness and DIF directly. First, they are parameterized differently: the original definition (2.1) is parameterized by a Lipschitz

constant $L$, while DIF is parameterized by an $(\epsilon, \delta)$ pair. Intuitively, (2.1) enforces (approximate) invariance at all scales (at any $\epsilon > 0$), while DIF only enforces invariance at one scale (determined by the input tolerance parameter). Second, (2.1) enforces invariance uniformly on $\mathcal{X}$, while (2.2) enforces invariance on average. Although DIF seems a weaker notion of algorithmic fairness than (2.1) (average fairness vs uniform fairness), DIF is actually more stringent in some ways because the constraints in (2.2) are looser. This is evident in the Mongé version of the optimization problem in (2.2):

$$\sup_{T:\mathcal{X}\to\mathcal{X}} \quad \mathbf{E}_P\big[d_\mathcal{Y}(h(X), h(T(X)))\big]$$
$$\text{subject to} \quad \mathbf{E}_P\big[d_\mathcal{X}(X, T(X))\big] \leq \epsilon. \tag{2.3}$$

The map corresponding to (2.1)

$$T_{\mathrm{IF}}(x) \triangleq \arg\max_{d_\mathcal{X}(x,x')\leq\epsilon} d_\mathcal{Y}(h(x), h(x'))$$

maps each $x$ to its worse-case counterpart $x'$ in (2.1). It is not hard to see that $T_{\mathrm{IF}}$ is a feasible map for (2.3), but it may not be optimal. This is because (2.3) only restricts $T$ to transport points by at most $\epsilon$ *on average*; the optimal $T$ may transport *some* points by more than $\epsilon$.

To make the two definitions more comparable, we consider an $\epsilon$-$\delta$ version of individual fairness. A model $h : \mathcal{X} \to \mathcal{Y}$ satisfies $(\epsilon, \delta)$-individual fairness at $x \in \mathcal{X}$ if and only if

$$d_\mathcal{Y}(h(x), h(T_{\mathrm{IF}}(x))) = \sup_{d_\mathcal{X}(x,x')\leq\epsilon} d_\mathcal{Y}(h(x), h(x')) \leq \delta. \tag{2.4}$$

To arrive at (2.4), we start by observing that (2.1) is equivalent to

$$\sup_{x\in\mathcal{X}} d_\mathcal{Y}(h(x), h(T_{\mathrm{IF}}(x))) \leq L\epsilon \text{ for any } x \in \mathcal{X} \text{ and } \epsilon > 0.$$

We fix $x$ and $\epsilon$ and re-parameterize the right side with $\delta$ to obtain (2.4). It is possible to show that if $h$ is $(\epsilon, \delta)$-DIF, then there exists $\delta'$ such that it satisfies $(\epsilon, \delta')$-individual fairness for "most" $x$'s. We formally state this result in a proposition.

**Proposition 2.3.** *If $h : \mathcal{X} \to \mathcal{Y}$ is $(\epsilon, \delta)$-DIF, then*

$$P_X(d_\mathcal{Y}(h(X), h(T_{\mathrm{IF}}(X))) \geq \tau) \leq \frac{\delta}{\tau} \text{ for any } \tau > 0.$$

## 2.3 ENFORCING DIF

There are two general approaches to enforcing invariance conditions such as (2.2). The first is distributionally robust optimization (DRO):

$$\min_{h\in\mathcal{H}} L_{\mathrm{adv}}(h) \triangleq \sup_{P':W_d(P,P')\leq\epsilon} \mathbf{E}_{P'}\big[\ell(Y', h(X'))\big], \tag{2.5}$$

where $\ell$ is a loss function and $W_d(P, Q)$ is the Wasserstein distance between distributions on $\mathcal{X} \times \mathcal{Y}$ induced by the transport cost function $c((x, y), (x'y')) \triangleq d_\mathcal{X}(x, x') + \infty \cdot \mathbf{1}\{y \neq y'\}$. This approach is very similar to adversarial training, and it was considered by Yurochkin et al. (2020) for enforcing (their modification of) individual fairness. In this paper, we consider a regularization approach to enforcing (2.2):

$$\min_{h\in\mathcal{H}} L(h) + \rho R(h), \quad L(h) \triangleq \mathbf{E}\big[\ell(Y, h(X))\big], \tag{2.6}$$

where $\rho > 0$ is a regularization parameter and the regularizer $R$ is defined in (2.2). An obvious advantage of the regularization approach is it allows the user to fine-tune the trade-off between goodness-of-fit and fairness by adjusting $\rho$ (see Figure 1; in Figure 3 of Appendix D we show the lack of such flexibility in the method of Yurochkin et al. (2020)). As we shall see, although the two approaches share many theoretical properties, we show in Section 4 that the regularization approach has superior empirical performance. We defer a more in-depth comparison between the two approaches to subsection 2.4.

At first blush, the regularized risk minimization problem (2.6) is not amenable to stochastic optimization because $R$ is not an expected value of a function of the training examples. Fortunately, by appealing to duality, it is possible to obtain a dual formulation of $R$ that is suitable for stochastic optimization.

**Theorem 2.4** (dual form of $R$). *If $d_\mathcal{Y}(h(x), h(x')) - \lambda d_\mathcal{X}(x, x')$ is continuous (in $(x, x')$) for any $\lambda \geq 0$, then*

$$R(h) = \inf_{\lambda\geq 0}\{\lambda\epsilon + \mathbf{E}_{P_X}\big[r_\lambda(h, X)\big]\}, \quad r_\lambda(h, X) \triangleq \sup_{x'\in\mathcal{X}}\{d_\mathcal{Y}(h(X), h(x')) - \lambda d_\mathcal{X}(X, x')\}.$$

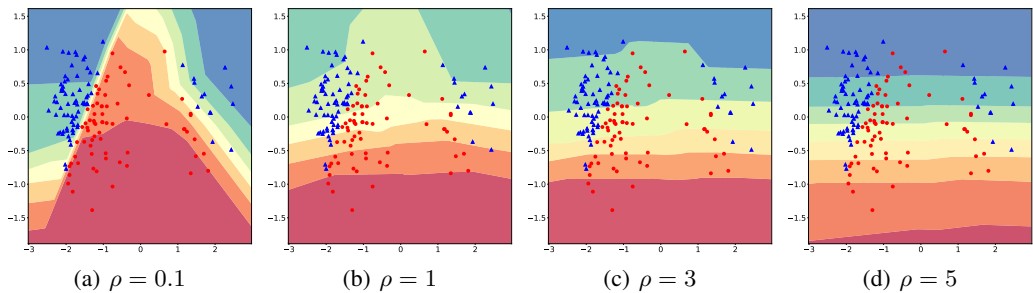

|(a) $\rho = 0.1$|(b) $\rho = 1$|(c) $\rho = 3$|(d) $\rho = 5$|

Figure 1: The decision surface of a one hidden layer neural network trained with SenSeI as the fair regularization parameter $\rho$ varies. In this ML task, points on a horizontal line (points with identical $y$-values) are similar, but the training data is biased because $P_{Y|X}$ is not constant on horizontal lines. We see that fair regularization (eventually) corrects the bias in the data.

We defer the proof of this result to Appendix A. In light of the dual form of the fair regularizer, the regularized risk minimization problem is equivalently

$$\min_{h \in \mathcal{H}} \inf_{\lambda \geq 0} \mathbf{E}_P\big[\ell(Y, h(X)) + \rho(\lambda \epsilon + r_\lambda(h, X))\big], \qquad (2.7)$$

where $r_\lambda$ is defined in Theorem 2.4, which has the form of minimizing an expected value of a function of the training examples. To optimize with respect to $h$, we parameterize the function space $\mathcal{H}$ with a parameter $\theta \in \Theta \subset \mathbf{R}^d$ and consider a stochastic approximation approach to finding the best parameter. Let $w \triangleq (\theta, \lambda)$ and $Z \triangleq (X, Y)$. The stochastic optimization problem we wish to solve is

$$\min_{w \in \Theta \times \mathbf{R}_+} F(w) \triangleq \mathbf{E}_P\big[f(w, Z)\big], \quad f(w, Z) \triangleq \ell(Y, h_\theta(X)) + \rho(\lambda \epsilon + r_\lambda(h_\theta, X)). \qquad (2.8)$$

It is not hard to see that (2.8) is a stochastic optimization problem. We summarize Sensitive Set Invariance (SenSeI) for stochastic optimization of (2.8) in Algorithm 1.

---

**Algorithm 1** SenSeI: Sensitive Set Invariance

---

**inputs:** starting point $(\widehat{\theta}_0, \widehat{\lambda}_0)$, step sizes $(\eta_t)$
**repeat**
$\quad (X_{t_1}, Y_{t_1}), \ldots, (X_{t_B}, Y_{t_B}) \sim P \qquad\qquad \triangleright$ sample mini-batch from $P$
$\quad x'_{t_b} \leftarrow \arg\max_{x' \in \mathcal{X}} \{d_{\mathcal{Y}}(h_{\widehat{\theta}_t}(X_{t_b}), h_{\widehat{\theta}_t}(x')) - \widehat{\lambda}_t d_{\mathcal{X}}(X_{t_b}, x')\}, b \in [B] \qquad \triangleright$ generate
worst-case examples
$\quad \widehat{\lambda}_{t+1} \leftarrow \max\{0, \widehat{\lambda}_t - \eta_t \rho(\epsilon - \frac{1}{B}\sum_{b=1}^B d_{\mathcal{X}}(X_{t_b}, x'_{t_b})\},$
$\quad \widehat{\theta}_{t+1} \leftarrow \widehat{\theta}_t - \eta_t(\frac{1}{B}\sum_{b=1}^B \partial_\theta\{\ell(Y_{t_b}, h_{\widehat{\theta}_t}(X_{t_b}))\} + \rho\partial_\theta\{d_{\mathcal{Y}}(h_{\widehat{\theta}_t}(X_{t_b}), h_{\widehat{\theta}_t}(x'_{t_b}))\})$
**until** converged

---

### 2.4 ADVERSARIAL TRAINING VS FAIR REGULARIZATION

Adversarial training is a popular approach to training invariant ML models. It was originally developed to defend ML models against adversarial examples (Goodfellow et al., 2014; Madry et al., 2017). There are many versions of adversarial training; the Wasserstein distributionally robust optimization (DRO) version by Sinha et al. (2017) is most closely related to (2.5). The direct goal of adversarial training is training ML models whose risk is small on adversarial examples, and the robust risk that adversarial training seeks to minimize (2.5) is exactly the risk on adversarial examples.

An indirect consequence of adversarial training is invariance to imperceptible changes to the inputs. Recall an adversarial example is a training example *with the same label as a non-adversarial training example* whose inputs differ imperceptibly from those of a training example. Thus (successful) adversarial training leads to ML models that ignore such imperceptible changes and are thus invariant in "imperceptible neighborhoods" of the training examples.

Unlike adversarial training, which leads to invariance as an indirect consequence of adversarial robustness, fair regularization enforces fairness by explicitly minimizing a fair regularizer. A key benefit of invariance regularization is it permits practitioners to fine-tune the trade-off between goodness-of-fit and invariance by adjusting the regularization parameter (see Figure 1).

## 2.5 RELATED WORK

There are three lines of work on enforcing individual fairness. There is a line of work that enforces group fairness with respect to many (possibly overlapping) groups to avoid disparate treatment of individuals (Hébert-Johnson et al., 2017; Kearns et al., 2017; Kim et al., 2018a;b). At a high-level, these methods repeatedly find groups in which group fairness is violated and updates the ML model to correct violations. Compared to these methods that approximate individual fairness with group fairness, we directly enforce individual fairness.

There is another line of work on enforcing individual fairness without knowledge of the similarity metric. Gillen et al. (2018); Rothblum & Yona (2018); Jung et al. (2019) reduce the problem of enforcing individual fairness to a supervised learning problem by minimizing the number of violations. Instead of a similarity metric, these algorithms rely on an oracle that detects violations of individual fairness. Garg et al. (2018) enforce individual fairness by penalizing the expected difference in predictions between counterfactual inputs. Instead of a similarity metric, this algorithm relies a way of generating counterfactual inputs. Our approach complements these methods by relying on a similarity metric instead of such oracles. This allows us to take advantage of recent work on learning fair metrics (Ilvento, 2019; Wang et al., 2019; Yurochkin et al., 2020; Mukherjee et al., 2020).

Most similar to our work is SenSR (Yurochkin et al., 2020) that also assumes access to a similarity metric. SenSR is based on adversarial training, i.e. it enforces a risk-based notion of individual fairness that only requires the risk of the ML model to be similar on similar inputs. In contrast, our method is based on fair regularization, i.e. it enforces a notion of individual fairness that requires the outputs of the ML model to be similar. The latter is stronger (it implies the former) and is much closer to Dwork et al.'s original definition. In addition, the risk-based notion of SenSR ties accuracy to fairness. Our approach separates these two (usually conflicting) goals and allows practitioners to more easily adjust the trade-off between accuracy and fairness as demonstrated in our experimental studies.

Finally, there is a line of work that proposes pre-processing techniques to learn a fair representation of the individuals and training an ML model that accepts the fair representation as input (Zemel et al., 2013; Bower et al., 2018; Madras et al., 2018; Lahoti et al., 2019). These works are complimentary to ours as we propose an in-processing algorithm.

## 3 THEORETICAL PROPERTIES OF SENSEI

In this section, we describe some theoretical properties of SenSeI. We defer all proofs to Appendix A. First, Algorithm 1 is an instance of a stochastic gradient method, and its convergence properties are well-studied. Even if $f(w, Z)$ is non-convex in $w$, the algorithm converges (globally) to a stationary point (see Appendix A for a rigorous statement). Second, the fair regularizer is data-dependent, and it is unclear whether minimizing its empirical counterpart (3.1) enforces distributional fairness. We show that the fair regularizer generalizes under standard conditions. Consequently,

1. it is possible for practitioners to *certify* that an ML model $h$ is DIF *a posteriori* by checking $\hat{R}(h)$ (even if $h$ was trained by minimizing $\widehat{R}$);
2. fair regularization enforces distributional individual fairness (as long as the hypothesis class includes DIF ML models);

**Notation** Let $\{(X_i, Y_i)\}_{i=1}^n$ be the training set and $\widehat{P}_X$ be the empirical distribution of the inputs. Define $\widehat{L} : \mathcal{H} \to \mathbf{R}$ as the empirical risk and $\widehat{R} : \mathcal{H} \to \mathbf{R}$ as the empirical counterpart of the fair regularizer $R$ (2.2):

$$\widehat{R}(h) \triangleq \left\{ \begin{array}{ll} \max_{\Pi \in \Delta(\mathcal{X} \times \mathcal{X})} & \mathbf{E}_\Pi\big[d_{\mathcal{Y}}(h(X), h(X'))\big] \\ \text{subject to} & \mathbf{E}_\Pi\big[d_{\mathcal{X}}(X, X')\big] \leq \epsilon \\ & \Pi(\cdot, \mathcal{X}) = \widehat{P}_X, \end{array} \right\}. \tag{3.1}$$

Define the loss class $\mathcal{L}$ and its counterpart for the fair regularizer as

$$\mathcal{L} \triangleq \{\ell_h : \mathcal{Z} \to \mathbf{R} \mid h \in \mathcal{H}\}, \quad \ell_h(z) \triangleq \ell(h(x), y),$$
$$\mathcal{D} \triangleq \{d_h : \mathcal{X} \times \mathcal{X} \to \mathbf{R}_+ \mid h \in \mathcal{H}\}, \quad d_h(x, x') \triangleq d_{\mathcal{Y}}(h(x), h(x')).$$

We measure the complexity of $\mathcal{D}$ and $\mathcal{L}$ with their entropy integrals with respect to to the uniform metric: $J(\mathcal{D}/\mathcal{L}) \triangleq \int_0^\infty \log N(\mathcal{D}/\mathcal{L}, \|\cdot\|_\infty, \epsilon)^{\frac{1}{2}} d\epsilon$, where $N(\mathcal{D}, \|\cdot\|_\infty, \epsilon)$ is the $\epsilon$-covering number of $\mathcal{D}$ in the uniform metric. The main benefit of using entropy integrals instead of Rademacher or Gaussian complexities to measure the complexity of $\mathcal{D}$ and $\mathcal{L}$ is it does not depend on the distribution of the training examples. This allows us to obtain generalization error bounds for counterfactual training sets that are similar to the (observed) training set. Finally, define the diameter of $\mathcal{X}$ in the $d_\mathcal{X}$ metric, that of $\mathcal{Y}$ in the $d_\mathcal{Y}$ metric as $D_\mathcal{X} \triangleq \sup_{x,x'\in\mathcal{X}} d_\mathcal{X}(x, x')$, $D_\mathcal{Y} \triangleq \sup_{y,y'\in\mathcal{Y}} d_\mathcal{Y}(y, y')$. The first result shows that the fair regularizer $R(h)$ generalizes. We assume $D_\mathcal{X}, D_{cY} < \infty$. This is a boundedness condition on $\mathcal{X} \times \mathcal{Y}$, and it is a common simplifying assumption in statistical learning theory. We also assume $J(\mathcal{D}) < \infty$. This is a standard assumption that appears in many uniform convergence results.

**Theorem 3.1.** *As long as $D_\mathcal{X}$, $D_\mathcal{Y}$, and $J(\mathcal{G})$ are all finite, with probability at least $1 - t$:*

$$\sup_{h\in\mathcal{H}} |\widehat{R}(h) - R(h)| \leq \frac{48(J(\mathcal{D}) + \frac{1}{\epsilon} D_\mathcal{X} D_\mathcal{Y})}{\sqrt{n}} + D_\mathcal{Y}(\frac{\log \frac{2}{t}}{2n})^{\frac{1}{2}}.$$

Theorem 3.1 implies it is possible to certify that an ML model $h$ satisfies distributional individual fairness (modulo error terms that vanish in the large-sample limit) by inspecting $\widehat{R}(h)$. This is important because a practitioner may inspect $\widehat{R}(h)$ after training to verify whether the trained ML model $h$ is fair enough. Theorem 3.1 assures the user that $\widehat{R}(h)$ is close to $R(h)$.

## 4 COMPUTATIONAL EXPERIMENTS

In this section we present empirical evidence that SenSeI trains individually fair ML models in practice and study the trade-off between accuracy and fairness parametrized by $\rho$ (defined in (2.6)).

**Baselines** We compare SenSeI to empirical risk minimization (Baseline) and two recent approaches for training individually fair ML models: Sensitive Subspace Robustness (SenSR) (Yurochkin et al., 2020) that uses DRO to achieve robustness to perturbations in a fair metric, and Counterfactual Logit Pairing (CLP) (Garg et al., 2018) that penalizes the differences in the output of an ML model on training examples and hand-crafted counterfactuals. We provide implementation details of SenSeI and the baselines in Appendix B.

### 4.1 TOXIC COMMENT DETECTION

We consider the task of training a classifier to identify toxic comments, *i.e.* rude or disrespectful messages in online conversations. Identifying and moderating toxic comments is crucial for facilitating inclusive online conversations. Data is available through the "Toxic Comment Classification Challenge" Kaggle competition. We utilize the subset of the dataset that is labeled with a range of identity contexts (*e.g.* "muslim", "white", "black", "homosexual gay or lesbian"). Many of the toxic comments in the train data also relate to these identities leading to a classifier with poor test performance on the sets of comments with some of the identity contexts (group fairness violation) and prediction rule utilizing words such as "gay" to flag a comment as toxic (individual fairness violation). To obtain good features we use last layer representation of BERT (base, uncased) (Devlin et al., 2018) fine tuned on a separate subset of 500k randomly selected comments without identity labels. We then train a 2000 hidden units neural network with these BERT features.

**Counterfactuals and fair metric.** CLP (Garg et al., 2018) requires defining a set of counterfactual tokens. The training proceeds by taking an input comment and if it contains a counterfactual token replacing it with another random counterfactual token. For example, if "gay" and "straight" are among the counterfactual tokens, comment "Some people are gay" may be modified to "Some people are straight". Then difference in logit outputs of the classifier on the original and modified comments is used as a regularizer. For toxicity classification Garg et al. (2018) adopted a set of 50 counterfactual tokens from (Dixon et al., 2018). Counterfactuals allow for a simple fair metric learning procedure via factor analysis: since any variation in representation of a data point and its counterfactuals is considered undesired, Yurochkin et al. (2020); Mukherjee et al. (2020) proposed to use a Mahalanobis metric with the major directions of variation among counterfactuals projected out. We utilize this

Table 1: Summary of **Toxicity** classification experiment over 10 restarts

|          | BA          | CTF score   | PC          | BA STD      | ACC STD     |
|----------|-------------|-------------|-------------|-------------|-------------|
| Baseline | **0.807**±0.002 | 0.072±0.002 | 0.621±0.014 | 0.044±0.003 | 0.081±0.004 |
| SenSeI   | 0.791±0.005 | **0.029**±0.005 | **0.773**±0.043 | **0.035**±0.003 | **0.052**±0.003 |
| SenSR    | 0.794±0.003 | 0.043±0.005 | 0.729±0.044 | 0.036±0.002 | 0.059±0.004 |
| CLP      | 0.795±0.006 | 0.032±0.006 | 0.763±0.048 | 0.038±0.003 | 0.056±0.004 |

Table 2: Summary of **Bios** classification experiment over 10 restarts

|          | BA          | CTF score   | PC          | Gap RMS     | Gap ABS     |
|----------|-------------|-------------|-------------|-------------|-------------|
| Baseline | 0.842±0.002 | 0.028±0.001 | 0.942±0.001 | 0.120±0.004 | 0.080±0.003 |
| SenSeI   | **0.843**±0.003 | **0.003**±0.000 | **0.977**±0.001 | **0.086**±0.005 | **0.054**±0.003 |
| SenSR    | 0.842±0.003 | 0.004±0.000 | 0.976±0.001 | 0.087±0.004 | **0.054**±0.003 |
| CLP      | 0.841±0.003 | 0.005±0.000 | 0.974±0.001 | 0.087±0.005 | 0.056±0.004 |

approach here to obtain fair metric for training SenSR (Yurochkin et al., 2020) and SenSeI. See Appendix B.1 for additional details regarding the fair metric construction in the experiments.

**Comparison metrics.** To evaluate individual fairness of the classifiers we use test data and 50 counterfactuals to check if the classifier toxicity decision varies across counterfactuals. For example, is prediction for "Some people are gay" same as for "Some people are straight"? An intuitive fair metric should be 0 on a pair of such comments and prediction of the classifier should not change based on the Dwork et al.'s individual fairness definition. We report Counterfactual Token Fairness (CTF) score (Garg et al., 2018) that quantifies variance across counterfactuals of the predicted probability that a comment is toxic, and Prediction Consistency (PC) equal to the portion of test comments where prediction is the same across *all* 50 counterfactual variations.

For goodness-of-fit we use balanced accuracy due to class imbalance. To quantify group fairness we follow accuracy parity notion (Zafar et al., 2017; Zhao et al., 2019). Here protected groups correspond to the identity context labels available in the data and accuracy parity quantifies whether a classifier is equally accurate on, e.g., comments labeled to have "white" context and those labeled with "black". There are 9 protected groups and we report standard deviation of the accuracies and balanced accuracies across them. We present mathematical expressions for each of the metrics in Appendix C for completeness.

**Results.** We repeat our experiment 10 times with random 70-30 train-test splits, every time utilizing a random subset of 25 counterfactuals during training. Results are summarized in Table 1. SenSeI on average outperforms other fair training methods on all individual and group fairness metrics at the cost of slightly lower balanced accuracy. SenSR has the lowest prediction consistency score suggesting that our fair regularization is more effective in enforcing individual fairness than adversarial training. This observation aligns with the empirical study by Yang et al. (2019) comparing various invariance enforcing techniques for spatial robustness in image recognition.

SenSeI also outperforms CLP: our approach uses optimization to find worst case perturbations of the data according to a fair metric induced by counterfactuals, while CLP chooses a random perturbation among the counterfactuals. SenSeI is searching for a perturbation on every data point that potentially allows to generalize to unseen counterfactuals, while CLP can only perturb comments that explicitly contain a counterfactual known during training time. In Figure 2 we verify that both SenSeI and CLP allow to "trade" fairness and accuracy by varying the regularization strength $\rho$ (in the table we used $\rho = 5$ for both). We also notice the effect of worst case optimization opposed to random sampling: SenSeI has higher prediction consistency at the cost of balanced accuracy.

## 4.2 OCCUPATION PREDICTION

Online professional presence via dedicated services or personal websites is the common practice in many industries. ML systems can be trained using data from these sources to identify person's occupation and used by recruiters to assist in finding suitable candidates for the job openings. Gender imbalances in occupations may trigger biases in such ML systems exacerbating societal inequality. De-Arteaga et al. (2019) proposed Bias in Bios dataset to study fairness in occupation prediction from

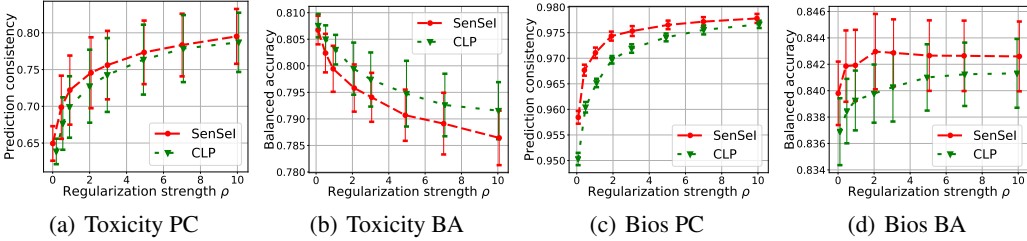

Figure 2: Balanced accuracy (BA) and prediction consistency (PC) trade-off.

Table 3: **Adult** experiment over 10 restarts. Prior methods are duplicated from Yurochkin et al. (2020)

|  | BA,% | S-Con. | GR-Con. | $\text{Gap}_G^{\text{RMS}}$ | $\text{Gap}_R^{\text{RMS}}$ | $\text{Gap}_G^{\text{max}}$ | $\text{Gap}_R^{\text{max}}$ |
|---|---|---|---|---|---|---|---|
| SenSR | 78.9 | .934 | .984 | .068 | .055 | .087 | .067 |
| Baseline | **82.9** | .848 | .865 | .179 | .089 | .216 | .105 |
| Project | 82.7 | .868 | **1.00** | .145 | .064 | .192 | .086 |
| Adv. debiasing | 81.5 | .807 | .841 | .082 | .070 | .110 | .078 |
| CoCL | 79.0 | - | - | .163 | .080 | .201 | .109 |
| SenSeI ($\rho = 40$) | 76.8 | **.945** | .963 | **.043** | **.054** | **.053** | **.064** |

a person's bio. The dataset consists of 400k textual bio descriptions and the goal is to predict one of the 28 occupations. We again use BERT fine-tuned on the train data to obtain bio representations and then train a 2000 hidden neurons neural networks using each of the fair training methods.

**Counterfactuals and fair metric.** Counterfactuals definition for this problem is the gender analog of the Bertrand & Mullainathan (2004) investigation of the racial bias in the labor market. For each bio we create a counterfactual bio by replacing male pronouns with the corresponding female ones and vice a versa. For the fair metric we use same approach as in the Toxicity study.

**Comparison metrics.** We use the same individual fairness metrics. To compare group fairness we report root mean squared gap (Gap RMS) and mean absolute gap (Gap ABS) between male and female true positive rates for each of the occupations following prior studies of this dataset (Romanov et al., 2019; Prost et al., 2019). For performance we report balanced accuracy due to imbalance in occupation proportions in the data.

**Results.** We repeat the experiment 10 times with 70-30 train-test splits and summarize results in Table 2 (for SenSeI and CLP we set $\rho = 5$). Comparing to the Toxicity experiment, we note that individual fairness metrics are much better. In particular, attaining prediction consistency is easier because there is only one type of counterfactuals. Overall, fairness metrics are comparable across fair training methods with a slight SenSeI advantage. It is interesting to note mild accuracy improvement: learning a classifier invariant to gender can help to avoid spurious correlations between occupations and gender present in the data. We present fairness accuracy "trade-off" in Figure 2 — increasing regularization strength has clear upward trend in terms of the prediction consistency without decreasing accuracy. This experiments is an example where fairness can be improved without "trading" performance.

We note two prior group fairness studies of the Bios dataset: Romanov et al. (2019) and Prost et al. (2019). Both reported worse classification results and fairness metrics. Better classification performance in our work is likely attributed to using BERT for obtaining bios feature vectors. For the group fairness, we note that relative to baseline improvement with SenSeI is more significant.

## 4.3 INCOME PREDICTION

The Adult dataset (Bache & Lichman, 2013) is a common benchmark in the group fairness literature. The task is to predict if a person earns more than $50k per year using information about their education, gender, race, marital status, hours worked per week, etc. Yurochkin et al. (2020) studied individual fairness on Adult by considering prediction consistency with respect to demographic features: race and gender (GR-Con.) and marital status (S-Con., i.e. spouse consistency). To quantify group fairness they used RMS gaps and maximum gaps between true positive rates across genders ($\text{Gap}_G^{\text{RMS}}$ and $\text{Gap}_G^{\text{max}}$) and races ($\text{Gap}_R^{\text{RMS}}$ and $\text{Gap}_R^{\text{max}}$). Due to class imbalance, performance is

quantified with balanced accuracy (B-Acc). For the fair metric they used Mahalanobis distance with race, gender and a logistic regression vector predicting gender projected out. We note that CLP was proposed as a fair training method for text classification (Garg et al., 2018) and is not applicable on Adult because it is not clear how to define counterfactuals. We compare SenSeI to results reported in Yurochkin et al. (2020). In Table 3 we show that with sufficiently large regularization strength $\rho = 40$ SenSeI is able to further reduce all group fairness gaps and improve one of the individual fairness metrics, however trading off some accuracy.

## 5 SUMMARY AND DISCUSSION

In this paper, we studied a regularization approach to enforcing individual fairness. We defined distributional individual fairness, a variant of Dwork et al.'s original definition of individual fairness and a data-dependent regularizer that enforces this distributional fairness (see Definition 2.1). We also developed a stochastic approximation algorithm to solve regularized empirical risk minimization problems and showed that it trains ML models with distributional fairness guarantees. Finally, we showed that the algorithm mitigates algorithmic bias on three ML tasks that are susceptible to such biases: income-level classification, occupation prediction, and toxic comment detection.

## ACKNOWLEDGEMENTS

This paper is based upon work supported by the National Science Foundation (NSF) under grants no. 1830247 and 1916271. Any opinions, findings, and conclusions or recommendations expressed in this paper are those of the authors and do not necessarily reflect the views of the NSF.

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

# A  THEORETICAL PROPERTIES

We collect the proofs of all the theoretical results in the paper here. We restate the results before proving them for the reader's convenience. We assume that $(\mathcal{X}, d_{\mathcal{X}})$ and $(\mathcal{Y}, d_{\mathcal{Y}})$ are complete and separable metric spaces (Polish spaces).

## A.1  DIF AND INDIVIDUAL FAIRNESS

**Proposition A.1** (Proposition 2.3)**.** *If* $h : \mathcal{X} \to \mathcal{Y}$ *is* $(\epsilon, \delta)$-*DIF, then*

$$P_X(\sup_{d_{\mathcal{X}}(x,x') \leq \epsilon} d_{\mathcal{Y}}(h(x), h(x')) \geq \tau) \leq \tfrac{\delta}{\tau}.$$

*Proof.* Recall

$$T_{\mathrm{IF}} \triangleq \arg\max_{d_{\mathcal{X}}(x,x') \leq \epsilon} d_{\mathcal{Y}}(h(x), h(x')).$$

is feasible for (the Mongé version of) (2.2). Thus $R(h) \leq \delta$ implies $\mathbf{E}_{P_X}\big[d_{\mathcal{Y}}(X, T_{\mathrm{IF}}(X))\big] \leq \delta$. Markov's inequality implies $P_X(d_{\mathcal{Y}}(X, T_{\mathrm{IF}}(X)) \geq \tau) \leq \frac{\delta}{\tau}$ for any $\tau > 0$. $\qquad\square$

## A.2  CONVERGENCE PROPERTIES OF SENSEI

Algorithm 1 is an instance of a stochastic gradient method, and its convergence properties are well-studied. Even if $f(w, Z)$ is non-convex in $w$, the algorithm converges (globally) to a stationary point. This a well-known result in stochastic approximation, and we state it here for completeness.

**Theorem A.2** (Ghadimi & Lan (2013))**.** *Let*

$$\sigma^2 \geq \mathbf{E}\big[\|\tfrac{1}{B}\sum_{b=1}^{B} \partial_w f(w, Z_b) - \partial F(w)\|_2^2\big]$$

*be an upper bound of the variance of the stochastic gradient. As long as* $F$ *is* $L$-*strongly smooth,*

$$F(w') \leq F(w) + \langle \partial F(w), w' - w \rangle + \tfrac{L}{2}\|w - w'\|_2^2$$

*for any* $w, w' \in \Theta \times \mathbf{R}_+$, *then Algorithm 1 with constant step sizes* $\eta_t = (\frac{2B\epsilon_0}{L\sigma^2 T})^{\frac{1}{2}}$ *satisfies*

$$\tfrac{1}{T}\sum_{t=1}^{T} \mathbf{E}\big[\|\partial F(w_t)\|_2^2\big] \leq \sigma(\tfrac{8L\epsilon_0}{BT}),$$

*where* $\epsilon_0$ *is any upper bound of the suboptimality of* $w_0$.

In other words, Algorithm 1 finds an $\epsilon$-stationary point of (2.8) in at most $O(\frac{1}{\epsilon^2})$ iterations. If $F$ has more structure (*e.g.* convexity), then Algorithm 1 may converge faster.

## A.3  PROOF OF DUALITY RESULTS IN SECTION 2

**Theorem A.3** (Theorem 2.4)**.** *If* $d_{\mathcal{Y}}(h(x), h(x')) - \lambda d_{\mathcal{X}}(x, x')$ *is continuous (in* $(x, x')$*) for any* $\lambda \geq 0$, *then*

$$R(h) = \inf_{\lambda \geq 0}\{\lambda\epsilon + \mathbf{E}_{P_X}\big[r_\lambda(h, X)\big]\},$$

$$r_\lambda(h, X) \triangleq \sup_{x' \in \mathcal{X}}\{d_{\mathcal{Y}}(h(X), h(x')) - \lambda d_{\mathcal{X}}(X, x')\}.$$

*Proof.* We abuse notation and denote the function $d_{\mathcal{Y}}(h(x), h(x'))$ as $d_{\mathcal{Y}} \circ h$. We recognize the optimization problem in (2.2) as an (infinite dimensional) linear optimization problem:

$$R(h) \triangleq \begin{cases} \sup_{\Pi:\Delta(\mathcal{X}\times\mathcal{X})} & \langle \Pi, d_{\mathcal{Y}} \circ h \rangle = \mathbf{E}_\Pi\big[d_{\mathcal{Y}}(h(X), h(X'))\big] \\ \text{subject to} & \langle \Pi, d_{\mathcal{X}} \rangle = \mathbf{E}_\Pi\big[d_{\mathcal{X}}(X, X')\big] \leq \epsilon \\ & \Pi(\cdot, \mathcal{X}) = P_X, \end{cases}$$

It is not hard check Slater's condition: $d\Pi(x, x') = \mathbf{1}\{x = x'\}dP(x)$ is strictly feasible. Thus we have strong duality (see Theorem 8.7.1 in (Luenberger, 1968)):

$$R(h) = \sup_{\Pi:\Pi(\cdot,\mathcal{X})=P_X} \inf_{\lambda \geq 0}\langle \Pi, d_{\mathcal{Y}} \circ h \rangle + \lambda(\epsilon - \langle \Pi, d_{\mathcal{X}} \rangle)$$

$$= \sup_{\Pi:\Pi(\cdot,\mathcal{X})=P_X} \inf_{\lambda \geq 0}\lambda\epsilon + \langle \Pi, d_{\mathcal{Y}} \circ h - \lambda d_{\mathcal{X}} \rangle$$

$$= \inf_{\lambda \geq 0}\{\lambda\epsilon + \sup_{\Pi:\Pi(\cdot,\mathcal{X})=P_X}\langle \Pi, d_{\mathcal{Y}} \circ h - \lambda d_{\mathcal{X}} \rangle\},$$

It remains to show

$$\sup_{\Pi:\Pi(\cdot,\mathcal{X})=P}\langle d_{\mathcal{Y}} \circ h - \lambda d_{\mathcal{X}}, \Pi \rangle = \mathbf{E}_P\big[\sup_{x' \in \mathcal{X}}\{d_{\mathcal{Y}}(h(X), h(x')) - \lambda d_{\mathcal{X}}(X, x')\}\big]. \quad \text{(A.1)}$$

$\leq$ **direction** The integrands in (A.1) satisfy

$$(d_{\mathcal{Y}} \circ h - \lambda d_{\mathcal{X}})(x, x') = d_{\mathcal{Y}}(h(x), h(x')) - \lambda d_{\mathcal{X}}(x, x') \leq \sup_{x' \in \mathcal{X}} \left\{ d_{\mathcal{Y}}(h(x), h(x')) - \lambda d_{\mathcal{X}}(x, x') \right\},$$

so the integrals satisfy the $\leq$ version of (A.1).

$\geq$ **direction** Let $\mathcal{Q}$ be the set of all Markov kernels from $\mathcal{X}$ to $\mathcal{X}$. We have

$$\sup_{\Pi : \Pi(\cdot, \mathcal{X}) = P} \langle d_{\mathcal{Y}} \circ h - \lambda d_{\mathcal{X}}, \Pi \rangle = \sup_{Q \in \mathcal{Q}} \int_{\mathcal{X} \times \mathcal{X}} d_{\mathcal{Y}}(h(x), h(x')) - \lambda d_{\mathcal{X}}(x, x') dQ(x' \mid x) dP(x)$$

$$\geq \sup_{T : \mathcal{X} \to \mathcal{X}} \int_{\mathcal{X}} d_{\mathcal{Y}}(h(x), h(T(x))) - \lambda d_{\mathcal{X}}(x, T(x)) dP(x),$$

where we recalled $Q(A \mid x) = \mathbf{1}\{T(x) \in A\}$ is a Markov kernel in the second step. (Technically, in the second step, we only sup over $T$'s that are *decomposable* (see Definition 14.59 in Rockafellar & Wets (2004)) with respect to $P$, but we gloss over this detail here.) We appeal to the technology of integrands (Rockafellar & Wets, 2004) to interchange integration and maximization. We assumed $d_{\mathcal{Y}} \circ h - \lambda d_{\mathcal{X}}$ is continuous, so it is a normal integrand (see Corollary 14.34 in Rockafellar & Wets (2004)). Thus it is OK to interchange integration and maximization (see Theorem 14.60 in Rockafellar & Wets (2004)):

$$\sup_{T : \mathcal{X} \to \mathcal{X}} \int_{\mathcal{X}} d_{\mathcal{Y}}(h(x), h(T(x))) - \lambda d_{\mathcal{X}}(x, T(x)) dP(x) = \int_{\mathcal{X}} \sup_{x' \in \mathcal{X}} \left\{ d_{\mathcal{Y}}(h(x), h(x')) - \lambda d_{\mathcal{X}}(x, x') \right\}, dP(x).$$

This shows the $\geq$ direction of (A.1). $\qquad\square$

We remark that it is not necessary to rely on the technology of normal integrands to interchange expectation and maximization in the proof of Theorem 2.4. For example, Blanchet & Murthy (2016) prove a similar strong duality result without resorting to normal integrands. We do so here to simplify the proof.

## A.4 PROOFS OF GENERALIZATION RESULTS IN SECTION 3

**Theorem A.4** (Theorem 3.1). *As long as $D_{\mathcal{X}}$, $D_{\mathcal{Y}}$, $J(\mathcal{G})$ are all finite,*

$$\sup_{f \in \mathcal{F}} |\mathbf{E}_{\widehat{P}}[f(Z)] - \mathbf{E}_P[f(Z)]| \leq \frac{48(J(\mathcal{D}) + \frac{1}{\epsilon} D_{\mathcal{X}} D_{\mathcal{Y}})}{\sqrt{n}} + D_{\mathcal{Y}} \left( \frac{\log \frac{2}{t}}{2n} \right)^{\frac{1}{2}}$$

*with probability at least $1 - t$.*

*Proof.* Let

$$\widehat{R}(h) \triangleq \begin{cases} \max_{\Pi \in \Delta(\mathcal{X} \times \mathcal{X})} & \mathbf{E}_{\Pi}[d_{\mathcal{Y}}(h(X), h(X'))] \\ \text{subject to} & \mathbf{E}_{\Pi}[d_{\mathcal{X}}(X, X')] \leq \epsilon \\ & \Pi(\cdot, \mathcal{X}) = \widehat{P}_X, \end{cases}.$$

where $\widehat{P}_X$ is the empirical distribution of the inputs in the training set. By Theorem 2.4, we have

$$\widehat{R}(h) - R(h) = \inf_{\lambda \geq 0} \{ \lambda \epsilon + \mathbf{E}_{\widehat{P}_X}[r_{\lambda}(h, X)] \} - \inf_{\lambda \geq 0} \{ \lambda \epsilon + \mathbf{E}_{P_X}[r_{\lambda}(h, X)] \}$$

$$= \inf_{\lambda \geq 0} \{ \lambda \epsilon + \mathbf{E}_{\widehat{P}_X}[r_{\lambda}(h, X)] \} - \lambda_* \epsilon - \mathbf{E}_{\widehat{P}_X}[r_{\lambda_*}(h, X)] \qquad (A.2)$$

$$\leq \mathbf{E}_{\widehat{P}_X}[r_{\lambda_*}(h, X)] \} - \mathbf{E}_{P_X}[r_{\lambda_*}(h, X)],$$

where $\lambda_* \in \arg\min_{\lambda \geq 0} \lambda \epsilon + \mathbf{E}_{P_X}[r_{\lambda}(h, X)]$. The infimum is attained because

$$\inf_{\lambda \geq 0} \{ \lambda \epsilon + \mathbf{E}_{P_X}[r_{\lambda}(h, X)] \}$$

is an strictly feasible (infinite-dimensional) linear optimization problem (see proof of Theorem 2.4). To bound $\lambda_*$, we observe that $r_{\lambda}(h, X) \geq 0$ for any $h \in \mathcal{H}$, $\lambda \geq 0$:

$$r_{\lambda}(h, X) = \sup_{x' \in \mathcal{X}} \{ d_{\mathcal{Y}}(h(X), h(x')) - \lambda d_{\mathcal{X}}(X, x') \}$$

$$\geq d_{\mathcal{Y}}(h(X), h(X)) - \lambda d_{\mathcal{X}}(X, X).$$

This implies

$$R(h) = \lambda_* \epsilon + \mathbf{E}_{P_X}\big[r_{\lambda_*}(h, X)\big] \geq \lambda_* \epsilon.$$

We rearrange to obtain a bound on $\lambda_*$:

$$\lambda_* \leq \frac{1}{\epsilon}R(h) \leq \frac{1}{\epsilon}D_{\mathcal{Y}} \triangleq \bar{\lambda}. \tag{A.3}$$

This is admittedly a crude bound, but it is good enough here. Similarly,

$$R(h) - \widehat{R}(h) \leq \mathbf{E}_{P_X}\big[r_{\widehat{\lambda}_*}(h, X)\big]\} - \mathbf{E}_{\widehat{P}_X}\big[r_{\widehat{\lambda}_*}(h, X)\big], \tag{A.4}$$

where $\widehat{\lambda}_* \in \arg\min_{\lambda \geq 0} \lambda\epsilon + \mathbf{E}_{\widehat{P}_X}\big[r_\lambda(h, X)\big]$, and $\widehat{\lambda}_* \leq \bar{\lambda}$. Combining (A.2) and (A.4), we obtain

$$|\widehat{R}(h) - R(h)| \leq \sup_{f \in \mathcal{F}} |\mathbf{E}_{\widehat{P}}\big[f(Z)\big] - \mathbf{E}_P\big[f(Z)\big]|,$$

where $\mathcal{F} \triangleq \{r_\lambda(h, \cdot) \mid h \in \mathcal{H}, \lambda \in [0, \bar{\lambda}]\}$. It is possible to bound $\sup_{f \in \mathcal{F}} |\mathbf{E}_{\widehat{P}}\big[f(Z)\big] - \mathbf{E}_P\big[f(Z)\big]|$ with results from statistical learning theory. First, we observe that the functions in $\mathcal{F}$ are bounded:

$$0 \leq r_\lambda(h, X) \leq \frac{1}{\epsilon}\sup_{y, y' \in \mathcal{Y}} d_{\mathcal{Y}}(y, y').$$

Thus $\sup_{f \in \mathcal{F}} |\mathbf{E}_{\widehat{P}}\big[f(Z)\big] - \mathbf{E}_P\big[f(Z)\big]|$ has bounded differences inequality, so it concentrates sharply around its expectation. By the bounded-differences inequality and a standard symmetrization argument,

$$\sup_{f \in \mathcal{F}} |\mathbf{E}_{\widehat{P}}\big[f(Z)\big] - \mathbf{E}_P\big[f(Z)\big]| \leq 2\Re_n(\mathcal{F}) + D_{\mathcal{Y}}(\frac{\log\frac{2}{t}}{2n})^{\frac{1}{2}}$$

with probability at least $1 - t$, where $\Re_n(\mathcal{F})$ is the Rademacher complexity of $\mathcal{F}$:

$$\Re_n(\mathcal{F}) = \mathbf{E}\big[\sup_{f \in \mathcal{F}} \frac{1}{n}\sum_{i=1}^n \sigma_i f(Z_i)\big].$$

It remains to study $\Re_n(\mathcal{F})$. First, we show that the $\mathcal{F}$-indexed Rademacher process $X_f \triangleq \frac{1}{n}\sum_{i=1}^n \sigma_i f(Z_i)$ is sub-Gaussian with respect to to the metric

$$d_{\mathcal{F}}((h_1, \lambda_1), (h_2, \lambda_2)) \triangleq \sup_{x_1, x_2 \in \mathcal{X}} |d_{\mathcal{Y}}(h_1(x_1), h_1(x_2)) - d_{\mathcal{Y}}(h_2(x_1), h_2(x_2))| + D_{\mathcal{X}}|\lambda_1 - \lambda_1| :$$

$$\begin{aligned}
&\mathbf{E}\big[\exp(t(X_{f_1} - X_{f_2}))\big] \\
&= \mathbf{E}\big[\exp\big(\frac{t}{n}\sum_{i=1}^n \sigma_i(r_{\lambda_1}(h_1, X_i) - r_{\lambda_2}(h_2, X_i))\big)\big] \\
&= \mathbf{E}\big[\exp\big(\frac{t}{n}\sigma(r_{\lambda_1}(h_1, X_i) - r_{\lambda_2}(h_2, X_i))\big)\big]^n \\
&= \mathbf{E}\big[\exp\big(\frac{t}{n}\sigma(\sup_{x_1' \in \mathcal{X}}\inf_{x_2' \in \mathcal{X}} d_{\mathcal{Y}}(h_1(X_i), h_1(x_1')) - \lambda_1 d_{\mathcal{X}}(x_1, X) - d_{\mathcal{Y}}(h_2(X_i), h_2(x_2')) + \lambda_2 d_{\mathcal{X}}(X, x_2))\big)\big]^n \\
&\leq \mathbf{E}\big[\exp\big(\frac{t}{n}\sigma(\sup_{x_1 \in \mathcal{X}} d_{\mathcal{Y}}(h_1(X_i), h_1(x_1')) - d_{\mathcal{Y}}(h_2(X_i), h_2(x_1')) + (\lambda_2 - \lambda_1)d_{\mathcal{X}}(x_1, X))\big)\big]^n \\
&\leq \exp\big(\tfrac{1}{2}t^2 d_{\mathcal{F}}(h_1, h_2)\big).
\end{aligned}$$

Let $N(\mathcal{F}, d_{\mathcal{F}}, \epsilon)$ be the $\epsilon$-covering number of $\mathcal{F}$ in the $d_{\mathcal{F}}$ metric. We observe

$$N(\mathcal{F}, d_{\mathcal{F}}, \epsilon) \leq N(\mathcal{D}, \|\cdot\|_\infty, \tfrac{\epsilon}{2}) \cdot N([0, \bar{\lambda}], |\cdot|, \tfrac{\epsilon}{2D_{\mathcal{X}}}). \tag{A.5}$$

By Dudley's entropy integral,

$$\begin{aligned}
\Re_n(\mathcal{F}) &\leq \frac{12}{\sqrt{n}}\int_0^\infty \log N(\mathcal{F}, d_{\mathcal{F}}, \epsilon)^{\frac{1}{2}} d\epsilon \\
&\leq \frac{12}{\sqrt{n}}\int_0^\infty \big(\log N(\mathcal{D}, \|\cdot\|_\infty, \tfrac{\epsilon}{2}) + N([0, \bar{\lambda}], |\cdot|, \tfrac{\epsilon}{2D_{\mathcal{X}}})\big)^{\frac{1}{2}} d\epsilon \\
&\leq \frac{12}{\sqrt{n}}\left(\int_0^\infty \log N(\mathcal{D}, \|\cdot\|_\infty, \tfrac{\epsilon}{2})^{\frac{1}{2}} d\epsilon + \int_0^\infty N([0, \bar{\lambda}], |\cdot|, \tfrac{\epsilon}{2D_{\mathcal{X}}})^{\frac{1}{2}} d\epsilon\right) \\
&\leq \frac{24J(\mathcal{D})}{\sqrt{n}} + \frac{24D_{\mathcal{X}}\bar{\lambda}}{\sqrt{n}}\int_0^{\frac{1}{2}}\log(\tfrac{1}{\epsilon})d\epsilon.
\end{aligned}$$

We check that $\int_0^{\frac{1}{2}}\log(\tfrac{1}{\epsilon})d\epsilon < 1$ to arrive at Theorem 3.1. $\qquad\square$

The chief technical novelty of this proof is the bound on $\lambda_*$ in terms of the diameter of the output space. This bound allows us to restrict the relevant function class in a way that allows us to appeal to standard techniques from empirical process theory to obtain uniform convergence results. In prior work (*e.g.* Lee & Raginsky (2017)), this bound relies on smoothness properties of the loss, but this precludes non-smooth $d_{\mathcal{Y}}$ in our problem setting.

**Corollary A.5.** *Assume there is $h_0 \in \mathcal{H}$ such that $L(h_0) + \rho R(h_0) < \delta_0$. As long as $D_{\mathcal{X}}$, $D_{\mathcal{Y}}$, $J(\mathcal{L})$, and $J(\mathcal{G})$ are all finite, any global minimizer $\hat{h} \in \arg\min_{h \in \mathcal{H}} \widehat{L}(h) + \rho \widehat{R}(h)$ satisfies*

$$L(\hat{h}) + \rho R(\hat{h}) \leq \delta_0 + 2\left(\frac{24J(\mathcal{L}) + 48\rho(J(\mathcal{D}) + \frac{1}{\epsilon}D_{\mathcal{X}}D_{\mathcal{Y}})}{\sqrt{n}} + (\bar{L} + \rho D_{\mathcal{Y}})(\frac{\log\frac{2}{t}}{2n})^{\frac{1}{2}}\right)$$

*with probability at least $1 - 2t$.*

*Proof.* Let $F(h) \triangleq L(h) + \rho R(h)$ and $\widehat{F}$ be its empirical counterpart. The optimality of $\hat{h}$ implies

$$F(\hat{h}) = F(\hat{h}) - \widehat{F}(\hat{h}) + \widehat{F}(\hat{h}) - \widehat{F}(h_0) + \widehat{F}(h_0) - F(h_0) + F(h_0) \leq \delta_0 + 2\sup_{h \in \mathcal{H}}|\widehat{F}(h) - F(h)|.$$

We have

$$\sup_{h \in \mathcal{H}}|\widehat{F}(h) - F(h)| \leq \sup_{h \in \mathcal{H}}|\widehat{L}(h) - L(h)| + \rho\sup_{h \in \mathcal{H}}|\widehat{R}(h) - R(h)|. \quad\quad (A.6)$$

We assumed $\ell$ is bounded, so $\sup_{h \in \mathcal{H}}|\widehat{L}(h) - L(h)|$ has bounded differences inequality, so it concentrates sharply around its expectation. By the bounded-differences inequality and a standard symmetrization argument,

$$\sup_{h \in \mathcal{H}}|\widehat{L}(h) - L(h)| \leq 2\mathfrak{R}_n(\mathcal{L}) + \bar{L}(\frac{\log\frac{2}{t}}{2n})^{\frac{1}{2}}$$

with probability at least $1 - t$, where $\mathfrak{R}_n(\mathcal{L})$ is the Rademacher complexity of $\mathcal{L}$. By Dudley's entropy integral,

$$\mathfrak{R}_n(\mathcal{L}) \leq \frac{12}{\sqrt{n}}\int_0^\infty \log N(\mathcal{L}, \|\cdot\|_\infty, \epsilon)^{\frac{1}{2}}d\epsilon,$$

so the first term on the right side of (A.6) is at most

$$\sup_{h \in \mathcal{H}}|\widehat{L}(h) - L(h)| \leq \frac{24J(\mathcal{L})}{\sqrt{n}} + \bar{L}(\frac{\log\frac{2}{t}}{2n})^{\frac{1}{2}}$$

with probability at least $1 - t$. Theorem 3.1 implies the second term on the right side of (A.6) is at most

$$\sup_{h \in \mathcal{H}}|\widehat{R}(h) - R(h)| \leq \frac{48(J(\mathcal{D}) + \frac{1}{\epsilon}D_{\mathcal{X}}D_{\mathcal{Y}})}{\sqrt{n}} + D_{\mathcal{Y}}(\frac{\log\frac{2}{t}}{2n})^{\frac{1}{2}}$$

with probability at least $1 - t$. We combine the bounds to arrive at the stated result. $\square$

## B  SENSEI AND BASELINES IMPLEMENTATION DETAILS

In this section we describe implementation details of all methods and hyperparameter selection to facilitate reproducibility of the experimental results reported in the main text.

**Improving balanced accuracy**   All three datasets we consider have noticeable class imbalances: over 80% comments in toxicity classification are non-toxic; several occupations in the Bias in Bios dataset are scarcely present (see Figure 1 in De-Arteaga et al. (2019) for details); about 75% of individuals in the Adult dataset make below $50k a year. Because of this class imbalance we choose to report balanced accuracy to quantify classification performance of different methods. Balanced accuracy is simply an average of true positive rates of all classes. To improve balanced accuracy for all methods we use balanced mini-batches following Yurochkin et al. (2020), i.e. when sampling a mini-batch we enforce that every class is equally represented.

**Fair regularizer distance metric**    Recall that fair regularizer in Definition 2.1 of the main text requires selecting a distance metric on the classifier outputs $d_{\mathcal{Y}}(h(x), h(x'))$. This distance is also required for the implementation of Counterfactual Logit Pairing (CLP) (Garg et al., 2018). In a $K$-class problem, let $h(x) \in \mathbf{R}^K$ denote a vector of $K$ logits of a classifier for an observation $x$, then for both SenSeI and CLP we define $d_{\mathcal{Y}}(h(x), h(x')) = \frac{1}{K}\|h(x) - h(x')\|_2^2$, i.e. mean squared difference between logits of $x$ and $x'$. This is one of the choices empirically studied by Yang et al. (2019) for image classification. We defer exploring alternative fairness regularizer distance metrics for future work.

**Data processing and classifier architecture**    Data processing and classifier are shared across all methods in all experiments. In Toxicity experiment we utilized BERT (Devlin et al., 2018) fine-tuned on a random 33% subset of the data. We downloaded the fine-tuned model from one of the Kaggle kernels.[1] In the Bios experiment for each train-test split we fine-tuned BERT-Base, Uncased[2] for 3 epochs with mini-batch size 32, learning rate $2e-5$ and 128 maximum sequence length. In both Toxicity and Bios experiments we obtained 768-dimensional sentence representations by average-pooling token embeddings of the corresponding fined-tuned BERTs. Then we trained a fully connected neural network with one hidden layer consisting of 2000 neurons with ReLU activations using BERT sentence representations as inputs.

For Adult experiment we followed data processing and classifier choice (i.e. 100 hidden units neural network) as described in Yurochkin et al. (2020).

**Hyperparameters selection**    In Table 4 for each hyperparameter we summarize its meaning, abbreviation, name in the code provided with the submission[3] and methods where it is used.

To select hyperparameters for each experiment we performed a grid search on an independent train-test split. Then we fixed selected hyperparameters and ran 10 experiment repetitions with random train test splits (these results are reported in the main text). Hyperparameter choices for all experiments are summarized in Tables 5, 6, 7. For the Adult experiment we duplicated results for all prior methods from Yurochkin et al. (2020).

Table 4: Hyperparameter names and notations

|  | notation | name in code | relevant methods |
|---|---|---|---|
| Number of optimization steps | $E$ | *epoch* | All |
| Mini-batch size | $B$ | *batch_size* | All |
| Parameter learning rate $\eta$ | $\eta$ | *lr* | All |
| Subspace attack step size | $s$ | *adv_step* | SenSeI, SenSR |
| Number of subspace attack steps | $se$ | *adv_epoch* | SenSeI, SenSR |
| Full attack step size | $f$ | *l2_attack* | SenSeI, SenSR |
| Number of full attack steps | $fe$ | *adv_epoch_full* | SenSeI, SenSR |
| Attack budget $\epsilon$ | $\epsilon$ | *ro* | SenSeI, SenSR |
| Fair regularization strength $\rho$ | $\rho$ | *fair_reg* | SenSeI, CLP |

Table 5: Hyperparameter choices in Toxicity experiment

|  | $E$ | $B$ | $\eta$ | $s$ | $se$ | $f$ | $fe$ | $\epsilon$ | $\rho$ |
|---|---|---|---|---|---|---|---|---|---|
| Baseline | 100k | 256 | 1e−5 | — | — | — | — | — | — |
| SenSR | 100k | 256 | 1e−5 | 0.1 | 10 | 0 | 0 | 0 | — |
| SenSeI | 100k | 256 | 1e−5 | 0.1 | 10 | 0 | 0 | 0 | 5 |
| CLP | 100k | 256 | 1e−5 | — | — | — | — | — | 5 |

---

[1] https://www.kaggle.com/taindow/bert-a-fine-tuning-example
[2] https://github.com/google-research/bert
[3] We will open-source the code and merge variable names with their abbreviations

Table 6: Hyperparameter choices in Bios experiment

|          | $E$  | $B$ | $\eta$ | $s$ | $se$ | $f$  | $fe$ | $\epsilon$ | $\rho$ |
|----------|------|-----|--------|-----|------|------|------|------------|--------|
| Baseline | 100k | 504 | 1e−6   | —   | —    | —    | —    | —          | —      |
| SenSR    | 100k | 504 | 1e−5   | 0.1 | 50   | 0.01 | 10   | 0.1        | —      |
| SenSeI   | 100k | 504 | 1e−5   | 0.1 | 50   | 0.01 | 10   | 0.1        | 5      |
| CLP      | 100k | 504 | 1e−5   | —   | —    | —    | —    | —          | 5      |

Table 7: Hyperparameter choices in Adult experiment

|        | $E$  | $B$  | $\eta$ | $s$ | $se$ | $f$   | $fe$ | $\epsilon$ | $\rho$ |
|--------|------|------|--------|-----|------|-------|------|------------|--------|
| SenSeI | 200k | 1000 | 1e−5   | 5   | 50   | 0.001 | 50   | 0.01       | 40     |

### B.1 FAIR METRIC LEARNING DETAILS

Following Yurochkin et al. (2020); Mukherjee et al. (2020) we consider the fair metric of the form $d_{\mathcal{X}}(x, x') = (x - x')^T \Sigma (x - x')$. We utilize their sensitive subspace idea writing $\Sigma = I - P_{ran(A)}$, i.e. an orthogonal complement projector of the subspace spanned by the columns of $A \in \mathbb{R}^{d \times k}$. Here $A$ encodes the $k$ directions of sensitive variations that should be ignored by the fair metric ($d$ is the data dimension), such as differences in sentence embeddings due to gender pronouns in the Bios experiment or due to identity (counterfactual) tokens in the Toxicity experiment.

**Synthetic experiment** In the synthetic experiment in Figure 1 we consider a fair metric ignoring variation along the $x$-axis coordinate, i.e. $A = [1\ \ 0]^T$.

**Toxicity experiment** To compute $A$ we utilize FACE algorithm of Mukherjee et al. (2020) (see section 2.1 and Algorithm 1 in their paper). Here groups of comparable samples are the BERT embeddings of sentences from the train data and their modifications obtained using 25 counterfactuals known at the training time. For example, suppose we have a sentence "Some people are gay" in the train data and the list of known counterfactuals is "gay", "straight" and "muslim". Then we can obtain two comparable sentences: "Some people are straight" and "Some people are muslim". BERT embeddings of the original and two created sentences constitute a group of comparable sentences. Embeddings of groups of comparable sentences are the inputs to Algorithm 1 of Mukherjee et al. (2020), which consists of a per-group centering step followed by a singular value decomposition. Taking the top $k = 25$ singular vectors gives us matrix of sensitive directions $A$ defining the fair metric.

**Bios experiment** We again utilize FACE algorithm of Mukherjee et al. (2020) to obtain the fair metric. Here the counterfactual modification is based on male and female gender pronouns. For example, sentence "He went to law school" is modified to "She went to law school". As a result, each group of comparable samples consists of a pair of bios (original and modified). Let $X \in \mathbb{R}^{n \times d}$ be the data matrix of BERT embeddings of the $n$ train bios, and let $X' \in \mathbb{R}^{n \times d}$ be the corresponding modified bios. Here Algorithm 1 of Mukherjee et al. (2020) is equivalent to performing SVD on $X - X'$. We take the top $k = 25$ singular vectors to obtain sensitive directions $A$ and the corresponding fair metric.

**Adult experiment** In this experiment $A$ consists of three vectors: a vector of zeros with 1 in the gender coordinate; a vector of zeros with 1 in the race coordinate; and a vector of logistic regression coefficients trained to predict gender using the remaining features (and 0 in the gender coordinate). This sensitive subspace construction replicates the approach Yurochkin et al. (2020) utilized in their Adult experiment for obtaining the fair metric. Please see Appendix B.1 and Appendix D in their paper for additional details.

## C  Fairness evaluation metrics definitions

**Individual fairness**   To compare individual fairness we used two metrics: prediction consistency and Counterfactual Token Fairness (CTF) score of Garg et al. (2018). The idea behind these metrics is to quantify changes in prediction when modifying original data in ways that intuitively should not change behavior of an individually fair classifier.

For Toxicity experiment an individually fair classifier should not change its prediction when a word "gay" in a comment is replaced with a word "straight". For example, we expect toxicity predictions on "Some people are gay" and "Some people are straight" to be the same. Following prior work (Dixon et al., 2018; Garg et al., 2018) we considered a set of 50 tokens[4] that should not affect the classifier when interchanged. For any comment that contains at least one of these 50 tokens we can create 49 versions of it via a simple word replacement and evaluate classifier prediction and probability of being toxic for each of the 50 variations (including the original). Prediction consistency is the proportion of comments (with at least one of the 50 tokens) where prediction is the same on *all* 50 variations. CTF score is the average (across all comments with at least one of the 50 tokens) standard deviation of the toxicity probability across 50 variations.

We use similar individual fairness metrics for the Bios experiment. We create a single variation of each bio by interchanging "he" and "she"; "his" and "her"; "him" and "hers"; "himself" and "herself"; "mr" and "ms" or "mrs"; original name with a random name from a different gender sampled among those present in the data. Prediction consistency is computed as before using 2 variations (including the original one) of each bio. Note that although there are fewer variations, there are significantly more classes in the Bios dataset. CTF score in the average (across all bios) squared Euclidean distance between the vectors of class probabilities for the 2 bio variations.

In the Adult experiment we compute same individual fairness metrics as in Yurochkin et al. (2020). S-Con. (spouse consistency) is the prediction consistency when creating data variations by altering marital status feature. GR-Con. (gender and race consistency) is the prediction consistency when creating data variations by altering race and gender features.

**Group fairness**   In our experiments we observed that enforcing individual fairness also has positive effect on group fairness metrics.

In the Toxicity experiment we used accuracy parity (Zafar et al., 2017; Zhao et al., 2019) to quantify group fairness. There are multiple protected groups in the Toxicity dataset (e.g. "muslim", "white", "black", "homosexual or lesbian") that correspond to human annotated identity contexts (not necessarily mutually exclusive). To account for this when evaluating accuracy parity we computed accuracies for each of the protected groups and reported their standard deviation. Large standard deviation implies that classifier is significantly more accurate on some protected groups than on the others. Because of the class imbalance we also reported standard deviation of the corresponding balanced accuracies.

For the Bios experiment we used same group fairness metrics as in the prior works studying this dataset (Romanov et al., 2019; Prost et al., 2019). Here protected attribute is binary: male or female genders. Let $\text{TPR}_{0,k}$ and $\text{TPR}_{1,k}$ denote true positive rates for class $k$ for protected attributes 0 and 1. Then TPR gap for class $k$ is $\text{Gap}_k = |\text{TPR}_{0,k} - \text{TPR}_{1,k}|$. The summary statistics we report are Gap RMS $= \sqrt{\frac{1}{K}\sum_k \text{Gap}_k^2}$ and Gap ABS $= \frac{1}{K}\sum_k \text{Gap}_k$.

For the Adult experiment we used same group fairness metrics as in Yurochkin et al. (2020), which correspond to Gap RMS described above and Gap MAX $= \max_k \text{Gap}_k$, evaluated with respect to race and gender (both are binary protected attributes in the dataset).

## D  Additional fairness-accuracy trade-off results

**Synthetic data experiment**   In Figure 1 we demonstrated how SenSeI allows to control fairness-accuracy trade-off in simulations by plotting the decision boundary of the corresponding classifier for

---

[4]https://github.com/conversationai/unintended-ml-bias-analysis/blob/master/unintended_ml_bias/bias_madlibs_data/adjectives_people.txt

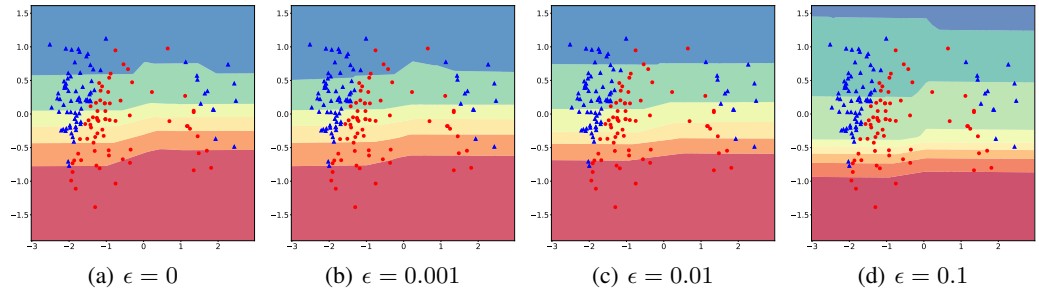

(a) $\epsilon = 0$     (b) $\epsilon = 0.001$     (c) $\epsilon = 0.01$     (d) $\epsilon = 0.1$

Figure 3: The decision surface of a one hidden layer neural network trained with SenSR (Yurochkin et al., 2020) as the DRO radius $\epsilon$ varies. Problem setting is the same as in Figure 1. Even for $\epsilon = 0$, SenSR prioritizes fairness over accuracy producing a horizontal decision surface. It is unable to achieve intermediate behaviors of SenSeI trading accuracy and fairness as in Figure 1 (a,b,c).

varying $\rho$. In Figure 3 we show the lack of such flexibility in SenSR (Yurochkin et al., 2020): varying the radius of the DRO ball $\epsilon$ in their definition of individual fairness results in a horizontal decision boundary even for $\epsilon = 0$. SenSR ties loss to fairness in its objective, and in this experiment loss can be increased significantly for anything but a horizontal decision boundary (fair metric allows free movement along the x-axis and a data-point can be perturbed in horizontal direction even for $\epsilon = 0$).

**Toxicity and Bios experiments** In Figure 2 we presented trade-offs between prediction consistency and balanced accuracy for SenSeI and CLP (Garg et al., 2018) for the Toxicity and Bios experiments. For completeness we also present corresponding CTF score and balanced accuracy trade-offs in Figure 4. As with the prediction consistency, we see that increasing fair regularization strength $\rho$ allows to train classifiers with better individual fairness properties. SenSeI outperforms CLP as it trains classifiers with lower CTF score across all values of $\rho$.

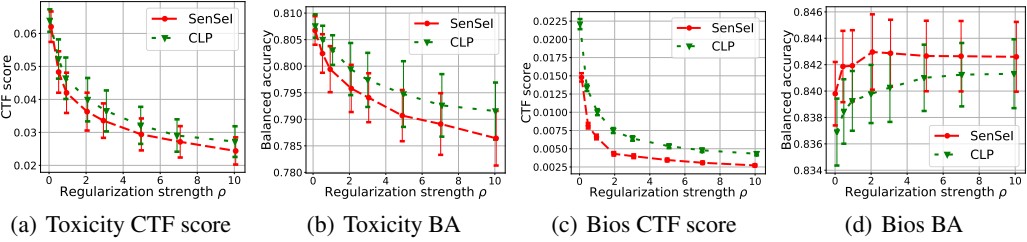

(a) Toxicity CTF score    (b) Toxicity BA    (c) Bios CTF score    (d) Bios BA

Figure 4: Balanced accuracy (BA) and CTF score trade-off on Toxicity and Bios experiments

