# OpenReview forum: "SenSeI: Sensitive Set Invariance for Enforcing Individual Fairness"
_ICLR.cc/2021/Conference — ICLR 2021 Oral_

### Official Review · AnonReviewer1 · 2020-10-27

**Rating:** 7
**Confidence:** 3

**Review:**

This paper proposes a variant of individual fairness, develops an algorithm to enforce this definition, and evaluates it. The fairness definition is based on transport, considering the maximum difference in outcomes for individuals drawn from a distribution "close" to the original data distribution in terms of a similarity metric.

The paper is well-written and thorough. The technical definition of individual fairness is a natural one, and it is amenable to stochastic optimization. Compared to other methods, SenSeI enforces the fairness constraint more strongly, at the cost of accuracy.

SenSeI comes with reasonably strong theoretical guarantees, which I appreciate.

The individual fairness constraints used in the experiments are all essentially counterfactual in nature -- certain pairs or groups of examples should get the same outcome, and there are no other constraints. Are there scenarios (even synthetic) where one could use more continuous similarity metrics?

This may be outside the scope of this work, but it would be interesting to dig deeper into which examples lead to fairness violations. For example, in the toxicity classifier, could an investigation into fairness violations tell us something about which counterfactuals are hard to treat equally, and therefore something about the way language is used?

---

> ### Author Response · Authors · 2020-11-19
> **Response to Reviewer 1**
>
> Thank you for your feedback. We provide answers to the questions below.
>
> **The individual fairness constraints used in the experiments are all essentially counterfactual in nature -- certain pairs or groups of examples should get the same outcome, and there are no other constraints. Are there scenarios (even synthetic) where one could use more continuous similarity metrics?**
>
> We note that for training SenSeI we always use continuous similarity metric (we have added additional details about the fair metric construction used for training to Appendix B.1). During evaluation we resort to counterfactual pairs/groups of examples for interpretability and following prior work on individual fairness (Garg et al. 2018, Yurochkin et al. 2020). In addition, in our synthetic experiment (Figure 1) we demonstrated that SenSeI learns decision boundary fair with respect to a continuous similarity metric. In this synthetic experiment fair metric is continuous and is equal to the distance along the y-axis (i.e. it ignores differences along the x-axis), therefore an individually fair classifier should assign same prediction to the points with the same y-axis values. As we increase the fair regularization strength $\rho$, SenSeI produces a neural network with horizontal decision boundary satisfying the aforementioned individual fairness constraint.
>
> **This may be outside the scope of this work, but it would be interesting to dig deeper into which examples lead to fairness violations. For example, in the toxicity classifier, could an investigation into fairness violations tell us something about which counterfactuals are hard to treat equally, and therefore something about the way language is used?**
>
> We agree that a more empirical study of the specific datasets would be interesting, but is outside the scope of our paper. When working on our experiments we explored the Bios dataset in a bit more detail. Some interesting observations are: replacing male pronouns with the corresponding female ones greatly increases performance of the baseline on male nurses at test time. This suggests that the baseline is learning to associate female gender pronouns with the nurse occupation, potentially due to strong gender imbalance within the nurse class. We also ran baseline occupation predictor on toy sentences such as "He went to law school" obtaining "attorney" as a prediction, but "She went to law school" resulted in a "paralegal" prediction. SenSeI corrects such issues, as measured by the prediction consistency, and improves performance on male nurses and other occupations with historic gender imbalance as measured by group fairness metrics.

---

### Official Review · AnonReviewer2 · 2020-10-28
**Well documented and theoretically-backed method for enforcing individual fairness.**

**Rating:** 7
**Confidence:** 4

**Review:**

############# Summary of contributions ##############

This paper tackles the problem of enforcing individual fairness. They define a notion of “distributional individual fairness” (DIF) which related to Dwork et al. 2011’s definition of individual fairness. They then provide an algorithm that enforces their DIF definition of individual fairness. They take as given the similarity metric required for evaluating individual fairness.

Specifically, contributions include:

- Theoretical: This paper proposes an “average case” definition for individual fairness which they call “distributional individual fairness” (DIF) (Definition 2.1). They theoretically compare DIF to the previous definition of individual fairness from Dwork et al. (Equation 2.1). DIF has the advantage of being computationally easier to enforce.

- Algorithmic: This paper proposes an algorithm for enforcing DIF using stochastic gradient methods.

- Theoretical: They provide a generalization bound for their algorithm (Theorem 3.1).

- Empirical: Using three real-world datasets, they evaluate their algorithm’s ability to satisfy both individual and group-based fairness constraints.

############# Strengths ##############

- The experiments are well set up and reflect real-world problems. The toxic comment detection experiment includes a counterfactual individual fairness metric that is reasonable and has precedent.

- Experiments evaluate group-based fairness metrics in addition to individual fairness metrics. These metrics are all relevant to their respective datasets and problem setups.

- The paper also discusses the theoretical relationship between their definition of “distributional individual fairness” (DIF) and the existing individual fairness definition proposed by Dwork et al. (Equation 2.1). This comparison is presented well.

- The paper provides theoretical analysis of the proposed algorithm. Theorem 3.1 provides a generalization bound for the SenSeI algorithm. Theorem 2.3 provides the dual form of the DIF regularizer which is used to solve their optimization problem for enforcing distributional individual fairness.

- The related work is well organized. It covers related work in enforcing individual fairness and learning similarity metrics.

- The hyperparameter selection procedure is well documented in Appendix B.

############# Weaknesses ##############

- It would be nice to have a clearer picture of the cases in which satisfying DIF actually leads to satisfying the individual fairness definition of Dwork et al. (Equation (2.1)), and vice versa. Currently, the theoretical comparison of DIF and Equation (2.1) feels a bit abstract and difficult to translate into real-world applications. Are there some distributional assumptions under which satisfying DIF implies satisfying Equation (2.1) (or vice versa)?

############# Recommendation ##############

Overall, my recommendation is 7: Good paper, accept. This paper provides a theoretically-backed algorithm for enforcing their definition of “distributionally individual fairness.” Their analysis is understandable, and their experiments appear detailed enough to be reproducible. My only major ask is for a more concrete and practically actionable description of the difference between DIF and Dwork et al.’s individual fairness.

############# Questions and clarifications ##############

- After Equation (2.3) on page 2, the authors state that the T(x) map corresponding to Equation (2.1) may not be an optimal point of Equation (2.3). Can the authors give a concrete example of a probability distribution P in Equation (2.3) when this T(x) is not optimal? This would give us further intuition for the difference between DIF (Definition 2.1) and Dwork et al.’s individual fairness (Equation (2.1)).

- The notation in Equation (2.4) is unclear. Is x’ a random variable with distribution P’, and if so, why is x’ lowercase (it seems that up to this point the paper has kept the convention that random variables are upper case and fixed values are lowercase)? What about the random variable Y? Is P’ the joint distribution of X’, Y’? If that’s the case, why does it say that W_{dX}(P, Q) is the Wasserstein distance between distributions on \mathcal{X}? It seems there are some typos here that the authors should clarify.

- In Definition 2.2, Is \Delta(\mathcal{X} \times \mathcal{X}) supposed to represent the set of all probability distributions over \mathcal{X} \times \mathcal{X}? I don’t see this defined explicitly anywhere.

- Is the \hat{h} notation really necessary at the top of page 6 in the last sentence before Section 4? The \hat{h} notation has not been defined or used up to this point. It seems that it would be sufficient to just use h in this sentence, as h has already been referred to as the trained ML model.

- Figure 1 is hard to read. The red/blue dots are not colorblind-friendly -- maybe you can use different symbols in addition to different colors, e.g. x’s for the red dots and o’s for the blue dots. The x and y axis labels are also extremely tiny.

---

> ### Author Response · Authors · 2020-11-19
> **Response to Reviewer 2**
>
> Thank you for the review. We address questions and concerns below. We have added a discussion regarding the connection between DIF and IF in subsection 2.2 and revised Figure 1 as you suggested.
>
> **It would be nice to have a clearer picture of the cases in which satisfying DIF actually leads to satisfying the individual fairness definition of Dwork et al., and vice versa.**
>
> Thank you for the suggestion of clarifying the relationship between DIF and Dwork et al's original definition of individual fairness (hereafter called IF for simplicity). We added a new subsection to section 2 (subsection 2.2) devoted solely to this topic. In summary, it is hard to compare DIF and IF directly (see the first paragraph of subsection 2.2), so we defined an $\epsilon$-$\delta$ version of IF and showed that DIF implies this $\epsilon$-$\delta$ version of IF "most of the time" (see Proposition 2.3).
>
> **Questions and clarifications**
>
> * The map $T(x)$ corresponding to (2.1) (now called $T_{IF}$ in the revised manuscript) is a feasible map for (2.3), but it may not be optimal for (2.3) because the optimal map for (2.3) *may* transport some points further than $\epsilon$. We elaborate on this in section 2.2 of the revised manuscript.
> * We fixed the typos around (2.4).
> * We clarified the notation $\Delta(\mathcal{X}\times\mathcal{X})$ in Definition 2.1.
> * We have removed the hats in the discussion at the end of section 3.
> * We have updated Figure 1 as you suggested.

---

### Official Review · AnonReviewer3 · 2020-10-29
**contribution is more tractable variant of Dwork et al’s “individual fairness" definition**

**Rating:** 7
**Confidence:** 2

**Review:**


Quality
- This paper is largely well-written with clearly stated theoretical results, and a range of experiments on three datasets.

Clarity
- What is the notation $\Delta(\mathcal{X} \times \mathcal{X})$?
- In the experiments section, SenSeI is compared with CLP, which is a method that uses hand-crafted counterfactuals. However, it appears that SenSeI’s implementation of individuals crucially requires hand-crafted counterfactuals as well? It was not clear to what the distance metric used in the experiments was. If the SenSeI implementation also uses the same counterfactuals, please comment on whether this is a fair comparison with CLP, and if other implementations of SenSeI (with different metrics) would yield the same results.
- Why are group fairness metrics compared in the experiments section? None of the methods studied target group fairness in general so the connection and relevance of group fairness should be elaborated upon.

Originality/Significance
- This paper’s original contribution is variant definition of Dwork et al’s “individual fairness” notion, that circumvents its computational intractability, by making it smooth. This is a significant enough contribution to the study of individual fairness, that makes it more applicable in practice.

---

> ### Author Response · Authors · 2020-11-19
> **Response to Reviewer 3**
>
> We thank the reviewer for the feedback. We have clarified the notation and added fair metric learning details in Appendix B.1, as you requested. We provide detailed answers to the clarity questions below.
>
> **What is the notation $\Delta(\mathcal X \times \mathcal X)$?**
>
> It is the set of probability measures on $\mathcal X \times \mathcal X$. We now define it inside Definition 2.1.
>
> **In the experiments section, SenSeI is compared with CLP, which is a method that uses hand-crafted counterfactuals. However, it appears that SenSeI’s implementation of individuals crucially requires hand-crafted counterfactuals as well? It was not clear to what the distance metric used in the experiments was. If the SenSeI implementation also uses the same counterfactuals, please comment on whether this is a fair comparison with CLP, and if other implementations of SenSeI (with different metrics) would yield the same results.**
>
> SenSeI does not require hand-crafted counterfactuals. Instead, it relies on a fair metric. When a counterfactual generating mechanism is available, it can be used to obtain a fair metric as in the Toxicity and Bios experiments. However, there could be situations where there is no clear way to generate counterfactuals, but it is still possible to learn a fair metric, as in the Adult experiment, where CLP is not applicable, but SenSeI is.
>
> It is true that in the Toxicity and Bios experiments we used the same counterfactuals for training CLP and for learning the fair metric to train SenSeI. The comparison is fair: both methods had access to the same amount of data/counterfactuals. SenSeI is more efficient due to its ability to find worst case perturbations of the data according to a fair metric induced by counterfactuals, while CLP simply chooses a random perturbation among the counterfactuals - please see an extended discussion in the last paragraph of Section 4.1. We have added details regarding the fair metric construction in Appendix B.1.
>
> **Why are group fairness metrics compared in the experiments section?**
>
> Group fairness metrics are presented for completeness, following prior work on individual fairness (e.g., Garg et al. 2018 and Yurochkin et al. 2020 also reported group fairness metrics). As you noted, the focus of this work is individual fairness, however empirically observing improvements of group fairness when enforcing individual fairness may be useful for future work studying connections between individual and group fairness.

---

### Official Review · AnonReviewer4 · 2020-10-29
**Good paper, the authors might use other reasons to motivate their paper.**

**Rating:** 7
**Confidence:** 3

**Review:**

This paper extends the individual fairness definition in order to (i) do statistical analysis (e.g., generalization bound) (ii) have the form of regularization so the practitioner can tune the parameter. Theorem 2.3 shows how to do stochastic optimization on the new definition, and Theorem 3.1 shows that that the new definition generalizes.

The paper is extremely well written, it was effortless to follow the arguments, and I enjoyed reading it. The theory part is sound and interesting, and the authors did a thorough experimental analysis on three datasets and compared their work with two baselines.


The authors motivate this work as a new formulation of individual fairness that decouples accuracy and fairness with a regularization term, enabling the practitioner to tune the parameter. I found this argument unconvincing. It is much easier for the practitioner to tune epsilon and delta, and the algorithm finds the minimum error while satisfying DIF. Tuning \ro after setting epsilon and delta is not very intuitive for practitioners.

Minor concerns:
In the related work, the authors argue that their work is different from previous extensions of the IF since they use the metric instead of the oracle. What is the big difference between using oracle and having access to metric? Can one serve instead of the other?
In the experiment section, the number reported for CLP is very different from their paper on comment toxicity detection.
I think having a few sentences on how to learn the individual metrics would be really good. (Right now, you refer to many work on learning metrics multiple times, but it’s never clear how they learn the similarity metric).
Comparing equation 2.1 and definition 2.1 is somewhat confusing. I think it would be more clear if you say 2.1 and 2.2 instead.

---

> ### Author Response · Authors · 2020-11-19
> **Response to Reviewer 4**
>
> We thank the reviewer for the feedback and provide answers to the raised concerns below. We have added additional details regarding the fair metric to Appendix B.1 as you requested.
>
> **The authors motivate this work as a new formulation of individual fairness that decouples accuracy and fairness with a regularization term, enabling the practitioner to tune the parameter. It is much easier for the practitioner to tune epsilon and delta, and the algorithm finds the minimum error while satisfying DIF.**
>
> Hyperparameter epsilon in the prior algorithm SenSR does not provide control over the fairness accuracy trade-off and its effect is harder to interpret for practitioners. In Figure 1 (simulations) and Figure 2 (real data) we showed how $\rho$ in SenSeI has a straightforward meaning of the accuracy-fairness trade-off. On the contrary, varying epsilon in SenSR does not have such effect. We added Figure 3 to the Appendix D visualizing decision boundary of SenSR for varying epsilon (analogous to Figure 1 for SenSeI) - even when epsilon is 0, SenSR prioritizes fairness over accuracy producing a horizontal decision surface. It is unable to achieve intermediate behaviors of SenSeI trading accuracy and fairness as in Figure 1 (a,b,c). Recall that the fair metric in this experiment allows free movement along the x-axis (i.e. a data point can be perturbed along the x-axis even when epsilon is 0). SenSR ties loss to fairness in its objective, and in this experiment loss can be increased significantly for anything but a horizontal decision boundary. This example demonstrates the advantage of SenSeI: decoupling fairness and accuracy allows explicit control of the trade-off via an additional hyperparameter $\rho$.
>
> **What is the big difference between using oracle and having access to metric? Can one serve instead of the other?**
>
> The fair metric and a fairness oracle are both ways in which the algorithm receives feedback regarding the fairness of its outputs, so they play similar roles in the fair training pipeline. Fair metrics are arguably easier to obtain in practice: there is prior work on learning fair metrics from data. On the other hand, fairness oracles typically requires human supervision during training. It is also possible to adapt a method that relies on a fair metric to use a fair oracle: we can appeal to metric learning methods to learn a fair metric that agrees with the oracle and train with this fair metric.
>
> **In the experiment section, the number reported for CLP is very different from their paper on comment toxicity detection.**
>
> CLP paper does not have an open-source code. We have reproduced the toxicity experiment and implemented their method to the best of our knowledge. We believe the results are quite similar. We report balanced accuracy to measure the performance. In their paper it can be eyeballed by looking at figure 2, i.e. (TPR + TNR)/2 is about 0.8, similar to the balanced accuracy of the CLP method reported in our paper. Regarding the CTF score (i.e. CTF gap), it is possible that our implementation differs from theirs (we are not sure what norm they meant when writing $|f(x) - f(x')|$ throughout the paper), however the idea of measuring variance in predicted probability across counterfactuals is preserved. Comparing to table 2 in their paper, our implementation of CLP has lower (better) CTF score and improvement over the baseline is more noticeable. We provide additional details regarding the CTF score and other fairness metrics used for comparison in Appendix C. Please let us know if there are any other differences between our and their results that we missed in this discussion.
>
> **Comparing equation 2.1 and definition 2.1 is somewhat confusing. I think it would be more clear if you say 2.1 and 2.2 instead.**
>
> We now say DIF/(2.2) instead of Definition 2.1 where necessary.

---

### Author Response · Authors · 2020-11-19
**General response**

We thank all reviewers for their time and thoughtful feedback. We address all questions individually and we have revised the paper accordingly. The main changes are: Subsection 2.2 comparing DIF and the original definition of IF; Appendix B.1 providing details regarding the fair metric learning in the experiments.

---

### Decision · Program_Chairs · 2021-01-07
**Final Decision**

**Decision:**

Accept (Oral)

**Comment:**

All of the reviewers agree that this paper is well-written, and provides sound theoretical analyses and comprehensive empirical evaluations. Overall, this paper makes a useful contribution in the direction of individual fairness. The authors have also addressed the concerns raised by the reviewers in their response.